# Predicting Complicated Appendicitis in Children: Pros and Cons of a New Score Combining Clinical Signs, Laboratory Values, and Ultrasound Images (CLU Score)

**DOI:** 10.3390/diagnostics13132275

**Published:** 2023-07-05

**Authors:** Konstantina Bekiaridou, Katerina Kambouri, Alexandra Giatromanolaki, Soultana Foutzitzi, Maria Kouroupi, Maria Aggelidou, Savas Deftereos

**Affiliations:** 1Department of Pediatric Surgery, Democritus University of Thrace, 68100 Alexandroupolis, Greece; bekiaridounadia@hotmail.com (K.B.); kampouri@med.duth.gr (K.K.); marartem@yahoo.gr (M.A.); 2Department of Pathology, Democritus University of Thrace, 68100 Alexandroupolis, Greece; agiatrom@med.duth.gr (A.G.); marykouroupi@gmail.com (M.K.); 3Department of Radiology, Democritus University of Thrace, 68100 Alexandroupolis, Greece; foutzita@gmail.com

**Keywords:** ultrasound, complicated appendicitis, uncomplicated appendicitis, score

## Abstract

Background: This retrospective study aimed to combine the clinical signs, laboratory values, and ultrasound images of 199 children with acute appendicitis in order to create a new predictive score for complicated appendicitis in children. Methods: The study included children who had clinical examination of abdominal pain (description of pain, anorexia, body temperature, nausea or vomiting, duration of symptoms), laboratory findings on admission (white blood cell, platelets, neutrophils, C-reactive protein), preoperative abdominal ultrasound, and histopathological report after an operation for appendicitis in their records during the period from January 2016 to February 2022. Results: According to the statistical analysis of the values using multivariate logistic regression models, the patients with appendiceal diameter ≥ 8.45 mm, no target sign appearance, appendicolith, abscess, peritonitis, neutrophils ≥ 78.95%, C-reactive protein ≥ 1.99 mg/dL, body temperature ≥ 38 °C, pain migration to right lower quadrant, and duration of symptoms < 24 h were more likely to suffer from complicated appendicitis. The new score was comprised of the 10 variables that were found statistically significant in the multivariate logistic model. Each of these variables was assigned a score of 1 due to the values that were associated with complicated appendicitis. Conclusions: A cutoff value of ≥4 has been a good indicator of the final score. The sensitivity with the usage of this score is 81.1%, the specificity 82.4%, the PPV 73.2%, the NPV approaches 88% and finally the accuracy is 81.9%. Also, the pros and cons of this score are discussed in this study.

## 1. Introduction

Acute appendicitis (AA) in the pediatric population is the most common reason of acute abdominal pain requiring surgical treatment. AA in children is still a diagnostic challenge for even experienced emergency physicians and pediatric surgeons [1,2]. The rate of errors in the primary diagnosis of AA ranges from 28% to 57% (children aged 2–12 years old) and is almost 100% in the ages <2 years, leading to an increased rate of complications, such as perforation, abscess, peritonitis, or sepsis [3,4]. Also, according to the literature, the percentage of negative appendectomies is stated as 3%, 10%, and sometimes 20% [2,5].

There is a tendency in recent years to not only establish a diagnosis of the AA, but to also distinguish preoperatively the uncomplicated cases of appendicitis from the complicated ones. The goal is either an early operation to prevent complications or conservative management to reduce the risk of a negative appendectomy [6]. Although the management of different forms of AA still remains controversial, many examinations including ultrasonography (US), computed tomography (CT) scan, magnetic resonance imaging (MRI), and diagnostic laparoscopy can be used to approach the right diagnosis [7]. In the pediatric population, US and CT are considered the gold standard methods for diagnosing AA, with radiologists and pediatric surgeons showing a particular preference for US due to the lack of radiation [8]. 

Until now, many scoring systems for AA have been proposed and designed to aid the clinical assessment of patients and reduce the rate of negative appendectomies [2,9]. Some studies in the past showed that the evaluation of preoperative total white blood cell count in combination with neutrophils and C-reactive protein could reduce negative appendectomy rates from 24% to 16% [10]. The Alvarado score, the Pediatric Appendicitis Score (PAS), and the Appendicitis Inflammatory Response (AIR) score have been most commonly used in the pediatric population [11,12,13]. The latter has been created not only to overcome the Alvarado score and PAS, but also to predict the severity of AA and the perforated appendicitis, including the C-reactive protein value (CRP) in the predictive score [2]. However, according to the literature, none of the aforementioned scoring systems may be used exclusively in establishing the diagnosis of AA in pediatric patients, and furthermore, some multicentric studies have evaluated the scores and found that none of the scores provides ideal PPV to be used in clinical practice as the method of choice for determination of the need for surgery [2,14,15,16,17].

In recent years, a few studies have attempted to combine the clinical, laboratory, and US findings in order to predict complicated appendicitis in the adult population [18,19]. However, in children, some studies have used the US and clinical findings to distinguish acute appendicitis from non-appendicitis [5,20,21] but only one is trying to predict complicated appendicitis using US, clinical, and laboratory findings [20].

The aim of this study was to evaluate a new predictive score (named CLU) that combines clinical signs, laboratory values and US images in order to differentiate pediatric uncomplicated (AUA) from complicated appendicitis (ACA) in children and discuss the pros and cons of this prediction.

## 2. Materials and Methods

In this study, which is a continuation of our previous study, we retrospectively examined the records of patients aged 0–14 years who were hospitalized in the Pediatric Surgery Department of Alexandroupolis University Hospital, Democritus University of Thrace, in order to design a new predictive score for complicated appendicitis [22]. The study included children who had clinical examination of abdominal pain (description of pain, anorexia, body temperature, nausea or vomiting, duration of symptoms) laboratory findings on admission (white blood cell (WBC), platelets (PLT), neutrophils (NEUT), C-reactive protein (CRP)) preoperative abdominal US, and histopathological report after an operation for AA (between January 2016 and February 2022). Exclusion criteria for our study were the absence of data related to these parameters in the child’s record, as well as cases of non-acute appendectomy, histopathologically confirmed normal appendix, and carcinoid or other pathology. In total, 52 out of 251 children were excluded from this study’s statistical analysis. Specifically, patients were excluded due to missing data (*n* = 14), non-acute appendectomy (*n* = 5), histopathologically confirmed carcinoid or other pathology (*n* = 3) non identified appendix in US (*n* = 25), and negative histopathology (*n* = 5). Methods of histopathological and US examination of our patients are well described in our previous study [22].

The study was conducted according to the guidelines of the Declaration of Helsinki, and the original protocol was approved by the Medical Ethics Committee of Alexandroupolis University Hospital (approval number 6809/19-02-2021).

Categorical variables were expressed as absolute and relative frequencies (*n*, %). Quantitative variables were presented as mean (±SD) values. All continuous variables were tested for normality using the Kolmogorov–Smirnov test. Hence, non-parametric Mann–Whitney U tests were applied to analyze differences between groups. A Pearson chi-square test was used for the comparisons of groups. A recipient-operator curve (ROC), with a calculation of sensitivity and specificity of the best cut-off and the area under the curve (AUC), were used to measure the diagnostic value of the continuous variables. Univariate and multivariate logistic regression analyses were applied to explore the sonographic findings, the symptoms, and laboratory findings associated with histopathological diagnosis of ACA. Odds ratios with 95% confidence intervals were computed from the results of the logistic regression analyses. For the evaluation of the predictive value of CLU, the receiver operating characteristic (ROC) curves were analysed. All statistical analyses were performed using IBM SPSS Statistics v25.0. The aforementioned statistical tests were performed at a 0.05 significance level.

## 3. Results

Demographic and clinical characteristics are presented in Table 1, which shows that 37.2% of the patients were diagnosed with ACA and 62.8% with AUA. The results showed that the majority of the sample were male (60.8%) with a mean (±SD) age 9.44 (±2.69) years. Patients with ACA were significantly younger than those with AUA (8.77 vs. 9.83, *p* = 0.031). The mean appendiceal diameter was significantly higher in patients with ACA (10.23 vs. 8.32, *p* < 0.001). A usual anatomical position was detected in the majority of the sample (89.9%). There was a significant difference between ACA and AUA patients in terms of distinct appendiceal wall layers, as the ACA patients had distinct appendiceal wall layers at a higher percentage (62.2% vs. 35.2%, *p* < 0.001). Of the patients, 93% had a non-compressible appendix. The majority of the patients with AUA had an appearance of target sign (70.4% vs. 41.9%, *p* < 0.001). No hypervascularisation (96%) and no lymphadenitis (65.8%) were observed in most patients. A higher percentage of ACA patients had appendicolith (41.9% vs. 13.6%, *p* < 0.001). Periappendiceal fat inflammation and free abdominal fluid were observed in most patients with ACA (73% vs. 49.6%, *p* = 0.001; 73% vs. 59.2%, *p* = 0.050, respectively). Diffuse free intraperitoneal fluid (DFIF) was detected at a higher percentage in the patients with ACA (20.3% vs. 9.6%, *p* = 0.034). Similar results were also found for free intraperitoneal fluid in the periappendiceal region (PFIF) (51.4% vs. 36%, *p* = 0.034). Free intraperitoneal fluid in Douglas’s pouch (DPFIF) was found in 24.1% of patients. Abscess (17.6% vs. 0.8%, *p* < 0.001) and peritonitis (16.2% vs. 1.6%, *p* < 0.001) were more common in patients with ACA. Statistically significant differences were also found in inflammatory laboratory markers, with the WBC count, NEUT, PLT, and CRP all being significantly higher in the group of ACA patients (*p* = 0.010, *p* < 0.001, *p* = 0.032, *p* < 0.001, respectively). RLQ tenderness to percussion, coughing, and hopping were observed in all ACA patients (100% vs. 50.4%, *p* < 0.001), and pain migration to the right lower quadrant (RLQ) was observed in most ACA patients compared to AUA (45.9% vs. 19.2%, *p* < 0.001). A significant higher percentage of ACA patients had anorexia (73% vs. 56.8%, *p* = 0.023), body temperature up to 38 °C (44.6% vs. 28%, *p* = 0.017), and nausea/vomiting (40.5% vs. 26.4%, *p* = 0.038). All patients had tenderness over right iliac fossa (RIF). Finally, 43.2% of patients had symptoms lasting 24–48 h. No statistically significant differences were found in terms of gender, anatomical position, non-compressible appendix, hypervascularisation, DPFIF, lymphadenitis, and duration of symptoms between the two groups. 

A cutoff value of ≥8.45 has proven to be a good indicator of the appendiceal diameter (mm). The corresponding values for the detection of acute complicated appendicitis, were 71.6% for sensitivity, 60% for specificity, PPV 51.5%, and NPV approaching 78.1% (AUC = 0.69; 95% CI: 0.609–0.769; *p* < 0.001) (Figure 1).

A cutoff value of ≥15.96 has proven to be a good indicator of the WBC. The corresponding values for the detection of the acute complicated appendicitis, were 51.4% for sensitivity, 61.6% for specificity, PPV 44.2%, and NPV approaching 68.1% (AUC = 0.61; 95% CI: 0.528–0.691; *p* = 0.010) (Figure 2).

A cutoff value of ≥78.95 has proven to be a good indicator of the ΝEUT. The corresponding values for the detection of the acute complicated appendicitis, were 70.3% for sensitivity, 51.2% for specificity, PPV 46%, and NPV approaching 74.4% (AUC = 0.64; 95% CI: 0.559–0.717; *p* < 0.001) (Figure 3).

A cutoff value of ≥321.50 has proven to be a good indicator of the PLT. The corresponding values for the detection of the acute complicated appendicitis, were 51.4% for sensitivity, 66.4% for specificity, PPV 47.5%, and NPV approaching 69.7% (AUC = 0.59; 95% CI: 0.506–0.676; *p* = 0.032) (Figure 4).

A cutoff value of ≥1.99 has proven to be a good indicator of the CRP. The corresponding values for the detection of the acute complicated appendicitis, were 77% for sensitivity, 58.4% for specificity, PPV 51.8%, and a NPV approaching 80.9% (AUC = 0.74; 95% CI: 0.666–0.811; *p* < 0.001) (Figure 5).

According to the multivariate logistic regression models, the patients with appendiceal diameter equal or higher than 8.45 (OR = 4.43, *p* = 0.009), no target sign appearance (OR = 4.12, *p* = 0.010), appendicolith (OR = 6.50, *p* = 0.001), abscess (OR = 40.05, *p* = 0.012), peritonitis (OR = 29.27, *p* = 0.048), NEUT equal or higher than 78.95 (OR = 3.48, *p* = 0.032), CRP equal or higher than 1.99 (OR = 3.46, *p* = 0.018), body temperature equal or higher than 38 °C (OR = 3.15, *p* = 0.038), pain migration to RLQ (OR = 4.17, *p* = 0.009), and duration of symptoms lower than 24 h (OR = 7.76, *p* = 0.015) were more likely to suffer from ACA (Table 2). The CLU score was comprised of the 10 variables that were found to be statistically significant in the multivariate logistic model. To construct the score, each of the above variables was assigned a value of 1 for those values that were associated with ACA. More specifically, if patients had an appendiceal appendix diameter equal or greater than 8.45, no appearance of target sign, appendicolith, abscess, peritonitis, NEUT equal or greater than 78.95, CRP equal or greater than 1.99, body temperature equal or greater than 38 °C, pain migration to the RLQ, and symptom duration less than 24 h, the CLU score was assigned a value of 10. 

A cutoff value of ≥4 has proven to be a good indicator of the final score. The corresponding values for the detection of the acute complicated appendicitis, were 81.1% for sensitivity, 82.4% for specificity, PPV 73.2%, a NPV approaching 88%, and an accuracy of 81.9% (AUC = 0.879; 95% CI: 0.830–0.928; *p* < 0.001) (Figure 6).

## 4. Discussion

The diagnosis of AA in children is not an easy task, especially in younger ones who often cannot describe their pain, or they present with nonspecific signs of abdominal pain. Doctors usually face a difficulty in deciding the course of treatment and the time of surgical intervention. Delaying the diagnosis of AA may be associated with increased recovery periods and hospitalization costs, risk of in-hospital infections, and higher morbidity and mortality [23,24]. 

The diagnostic pathway for acute abdominal pain in the emergency department of hospitals varies and mostly depends on doctor’s clinical experience. Recent studies tried to create algorithms to approach the right diagnosis, and finally, in 2015, the World Society of Emergency Surgery (WSES) organized in Jerusalem the first consensus conference producing evidence-based guidelines for the diagnosis and treatment of AA in adult patients, and in 2020 they updated the guidelines for adult and pediatric populations. In these studies, the usefulness of scores for the diagnosis of AA is discussed, but they recommend not making the diagnosis based on the already known scores alone, especially in children [25,26,27].

Recent studies try to differentiate preoperatively AUA from ACA, since the treatment for AUA is safe and can be non-operative, while the treatment of ACA is more complicated, especially in children younger than three years, as it is reported that the perforation rate of acute appendicitis is 80–100% for them, while it is approximately 38% in older children [28]. The accurate diagnosis of AA has been improved by using various scores [14]. However, it is still a challenge, especially in children, to predict preoperatively complicated appendicitis in order to decide the right management.

In the present retrospective study, a new score combining clinical, laboratory, and US findings is proposed to preoperatively distinguish AUA from ACA in children. This score comprised three clinical (body temperature equal or higher than 38 °C, pain migration to RLQ, duration of symptoms lower than 24 h), two laboratories (NEUT% equal or higher than 78.95 and CRP equal or higher than 1.99), and five US (appendiceal diameter equal or higher than 8.45 mm, presence of an appendicolith, no target sign appearance, peritonitis, and abscess) findings. A CLU ≥ 4 yielded an accuracy of 81.9%, a PPV of 73.2% and a NPV of 88% to predict complicated appendicitis, with sensitivity and specificity reaching 81.1% and 82.4% respectively. All of these parameters selected for the CLU score are easily accessible in daily practice. To our knowledge, this score is one of the two scores related to children using ultrasound, clinical, and laboratory findings in order to distinguish AUA from ACA. The other one was designed by Hao et al., who combined the US findings with the PAS score and found that the combination raised the specificity of ultrasound and PAS score relative to if they were calculated individually [20]. There is a third study in which a scoring system is made to diagnose AA from non-appendicitis in children using imaging-laboratory and clinical criteria, but in this study, there is not any concern for the diagnosis of AUA and ACA [5]. We created this score in order to achieve greater accuracy in the preoperative differentiation of AUA from ACA after combining more parameters. 

Surprisingly, and although statistically significant differences were found in WBC and PLT values, which were higher in the ACA patient group individually, and even though they are used as part of many appendicitis scores in the literature [2,29,30,31], they were excluded in the multivariate analysis and the CLU calculation of this study. This could be partly justified by the fact that laboratory markers have limited diagnostic utility by themselves because they are elevated in many infectious diseases, especially in children. In other studies, it is also reported that during the progression of inflammation of AA, the WBC count decreases after an initial higher value than the normal limits, so there is a possibility many of the children in this study with ACA would have WBC within normal limits [32]. Also, in our study, the ACA group contained many younger children, and it is widely accepted that the classical laboratory findings that seem to be the rule in older children or in adolescents are missing in younger children [3,30,33]. 

Other laboratory markers such as CRP seem to be useful for the diagnosis of AA, as CRP levels increase rapidly in the acute phase of inflammation. CRP alone as a marker does not have a high accuracy for diagnosing AA, but in combination with other markers its accuracy is much greater. Also, a low CRP value should be explained with caution if symptoms have only developed recently, as it seems to increase after 10–12 h of the initial symptoms [32,34,35]. In the present study, a CRP value equal or higher than 1.99 was statistically significant enough to be in the multivariate analysis of the model and became one of the indicators that make up the CLU score. In recent years, other laboratory markers have begun to be investigated, however, the published results are ambiguous. One of these is hyponatremia as a predictor marker for ACA [28]. Although this marker is easy to measure, we did not include it in our study because inflammation, dehydration, vomiting, and diarrhea, which may be present in other diseases besides AA, may cause sodium chloride deficiency.

Many of our patients (42.8%) had symptoms lasting 24–48 h. However, although someone could believe that the inflammatory response in AA is progressive (the longer the duration of symptoms the worse the severity of AA) [36], in our multivariate analysis it was observed that patients with duration of symptoms less than 24 h were more likely to suffer from ACA, hence why this was included in our new score. This result could be explained by the fact that in small children AA is an uncommon disease that has a varied presentation and complications that can develop rapidly, and in our study our patients with ACA were statistically younger than the patients with AUA [37]. Also, in our study, the ACA patients not only had perforated, gangrenous appendicitis or diffuse peritonitis, but also the majority had an appendicolith. Appendicolith in AA is referred as an independent prognostic risk factor although it is associated with appendiceal perforation. In recent literature, appendicolith appendicitis seems to have similar histopathological lesions as ACA, but most of the children with appendicolith appendicitis were associated with a shorter duration of symptoms [38]. In our study, in the ACA group there was a significant number of patients with appendicolith appendicitis (*n* = 13 in total *n* = 31 patients) with shorter duration of symptoms.

Many hospitals, especially in adult patients, rely on CT to diagnose appendicitis, as CT has a sensitivity ranged from 88 to 100% for adults and about 95% for the pediatric population. However, the use of CT for diagnosing ACA has lower sensitivity and specificity (62–81% respectively). A recent study describes that the combination of CT with laboratory and clinical findings increased the accuracy of CT for ACA [28,39]. However, the possibility of the future development of malignancy due to the radiation is the major problem with the use of CT as a diagnostic method in children. US is the method of choice for this condition, as it avoids radiation exposure, but its accuracy is widely varied as it depends on the operator’s training and experience [5]. Even in a center with well-trained radiologists, there is a low sensitivity when using ultrasound to diagnose complications of the appendix. According to Carpenter’s research, US has a sensitivity of 44.0%, a specificity of 93.1%, a PPV of 74.8%, and a NPV of 78.1% in diagnosing perforated appendicitis [40]. Ιn our study, the problem with the operator’s diagnostic ability has been overcome since, in the recognition of AA, a specific imaging approach protocol has been created by a specialist pediatric radiologist so that specific structures are sought during imaging by every operator. According to this, all US operators should work on classical real-time with gradually elevated compression US [41]. The US protocol started by “looking” at the whole abdominal cavity (free fluid and/or other pathology) leaving the right lower quadrant (RLQ) as the last part for evaluation (to avoid early onset of irritability). In this latter position (RLQ), graded compression was gently applied in order to gradually display the bowel loops and reveal the compressibility of the appendix. This compressibility or non-compressibility of the appendix was the major direct sign of normal appendix (or even when perforation could be present) or acute appendicitis, respectively. The maximum diameter of the appendix (normal < 6 mm) and the wall thickness (normal < 3 mm) were recorded and the presence of appendicolith (hyperechoic area with posterior shadowing) were evaluated. It was crucial to attempt to highlight the well-known target sign (hypoechoic fluid-filled lumen, hyperechoic mucosa/submucosa, and hypoechoic muscularis layer). Finally, images using colour Doppler US were obtained (hypervascularity in early stages of acute appendicitis, hypo- to avascularity in abscess and necrosis). This step-by-step approach can be followed in all examinations in order to record the so-called direct signs of acute appendicitis. Furthermore, indirect signs were recorded. These signs were free fluid surrounding the appendix, local abscess formation, increased echogenicity and/or uncompressible local mesenteric fat, enlarged local mesenteric lymph nodes, signs of secondary small bowel obstruction, and thickening of the peritoneum [22]. However, in the present study, all cases where the appendix was not recognized were removed. 

The major strength of our study is that it is one of only two studies in the literature where the score was designed to distinguish complicated from simple appendicitis in children by combining clinical, laboratory, and ultrasound findings. However, there were some limitations that could be mentioned. Although in our hospital there is an imaging protocol for appendicitis that every radiologist must apply, there were 25 (9.9%) cases where the appendix was not recognized, and these were excluded from our retrospective study. Also, more cases and studies are needed in order to calibrate this score. 

## 5. Conclusions

A new score for distinguishing ACA from AUA in pediatric patients with abdominal pain was designed. It combines clinical, laboratory, and US findings in order to increase US accuracy in those cases. The proposed score (named CLU) is simple to understand and remember. Its parameters are easily requested and it is cost-effective. It helps not only to diagnose AA, but it may also indicate the course of treatment for AA, as with this score someone can distinguish the ACA. With a value equal or greater than 4, its accuracy reaches 81.9% and a NPV of 88%.

More studies could be organized according to this score in order to calibrate the method and increase its accuracy.

## Figures and Tables

**Figure 1 diagnostics-13-02275-f001:**
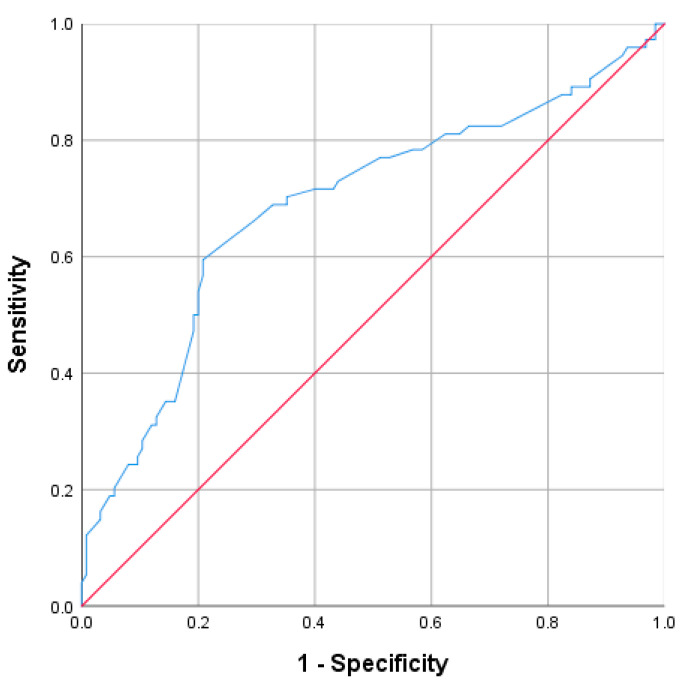
Receiver operating characteristic curve (ROC) for the appendiceal diameter (mm).

**Figure 2 diagnostics-13-02275-f002:**
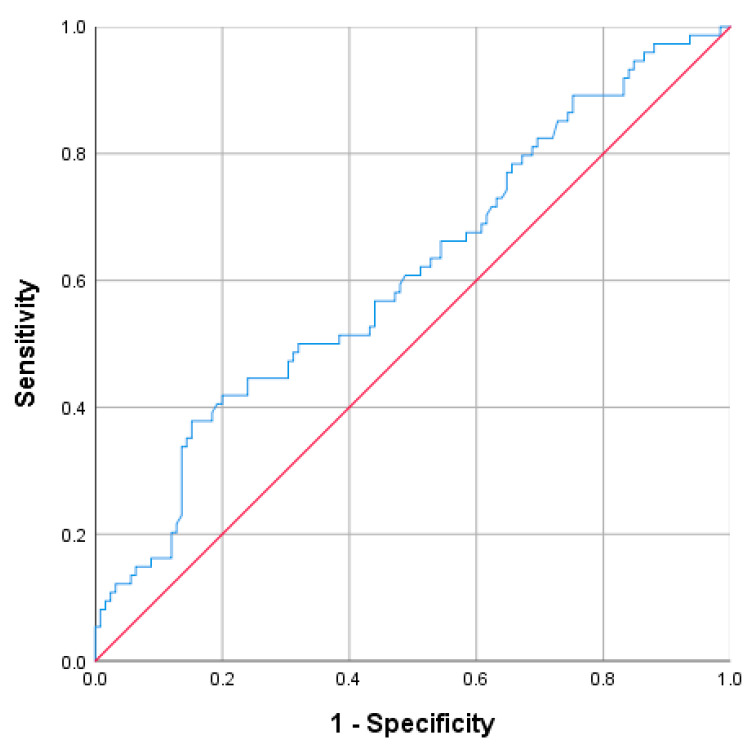
Receiver operating characteristic curve (ROC) for the WBC.

**Figure 3 diagnostics-13-02275-f003:**
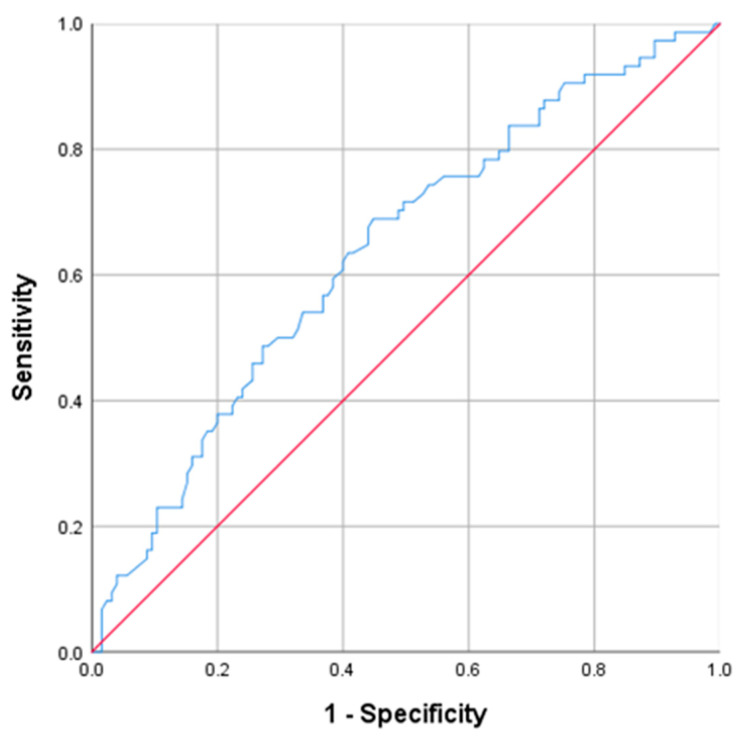
Receiver operating characteristic curve (ROC) for the ΝEUT.

**Figure 4 diagnostics-13-02275-f004:**
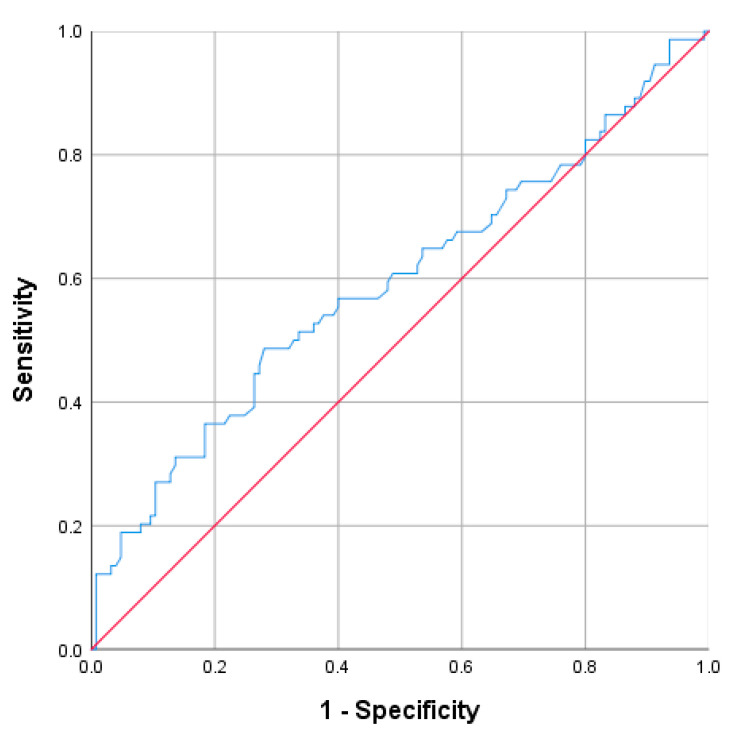
Receiver operating characteristic curve (ROC) for the PLT.

**Figure 5 diagnostics-13-02275-f005:**
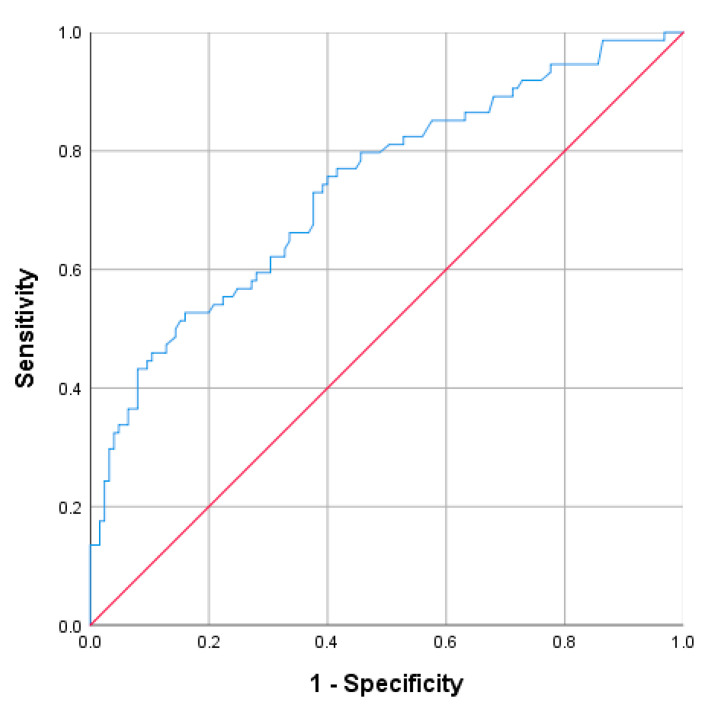
Receiver operating characteristic curve (ROC) for the CRP.

**Figure 6 diagnostics-13-02275-f006:**
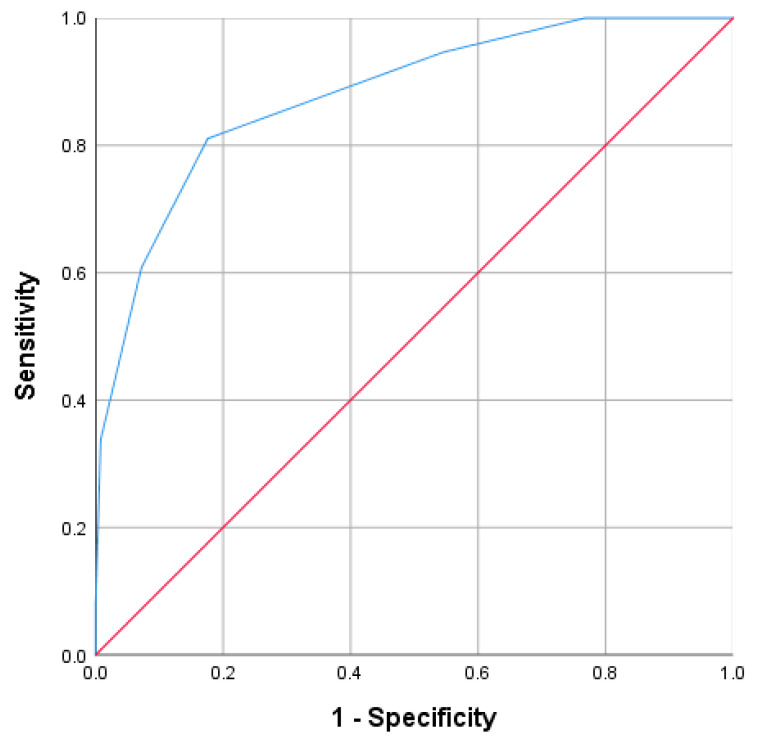
Receiver operating characteristic curve (ROC) for the final score.

**Table 1 diagnostics-13-02275-t001:** Patient characteristics by histopathological classification.

Characteristic	Total (*n* = 199)	ACA (*n* = 74)	AUA (*n* = 125)	*p*
**Age (years)**	9.44 (±2.69)	8.77 (±2.98)	9.83 (±2.44)	**0.031**
**Gender**				0.763
Male	121 (60.8%)	46 (62.2%)	75 (60%)	
Female	78 (39.2%)	28 (37.8%)	50 (40%)	
**Appendiceal diameter (mm)**	9.03 (2.87)	10.23 (3.51)	8.32 (2.13)	**<0.001**
**Anatomical position**				0.094
Usual	179 (89.9%)	70 (94.6%)	109 (87.2%)	
Unusual	20 (10.1%)	4 (5.4%)	16 (12.8%)	
**Distinct appendiceal wall layers**				**<0.001**
Yes	90 (45.2%)	46 (62.2%)	44 (35.2%)	
No	109 (54.8%)	28 (37.8%)	81 (64.8%)	
**Non-compressible**				0.489
Yes	185 (93%)	70 (94.6%)	115 (92%)	
No	14 (7%)	4 (5.4%)	10 (8%)	
**Target sign appearance**				**<0.001**
Yes	119 (59.8%)	31 (41.9%)	88 (70.4%)	
No	80 (40.2%)	43 (58.1%)	37 (29.6%)	
**Hypervascularisation**				0.140
Yes	8 (4%)	1 (1.4%)	7 (5.6%)	
No	191 (96%)	73 (98.6%)	118 (94.4%)	
**Appendicolith**				**<0.001**
Yes	48 (24.1%)	31 (41.9%)	17 (13.6%)	
No	151 (75.9%)	43 (58.1%)	108 (86.4%)	
**Periappendiceal fat inflammation**				**0.001**
Yes	116 (58.3%)	54 (73%)	62 (49.6%)	
No	83 (41.7%)	20 (27%)	63 (50.4%)	
**Free abdominal fluid**				**0.050**
Yes	128 (64.3%)	54 (73%)	74 (59.2%)	
No	71 (35.7%)	20 (27%)	51 (40.8%)	
**DFIF**				**0.034**
Yes	27 (13.6%)	15 (20.3%)	12 (9.6%)	
No	172 (86.4%)	59 (79.7%)	113 (90.4%)	
**PFIF**				**0.034**
Yes	83 (41.7%)	38 (51.4%)	45 (36%)	
No	116 (58.3%)	36 (48.6%)	80 (64%)	
**DPFIF**				0.329
Yes	48 (24.1%)	15 (20.3%)	33 (26.4%)	
No	151 (75.9%)	59 (79.7%)	92 (73.6%)	
**Lymphadenitis**				0.480
Yes	68 (34.2%)	23 (31.1%)	45 (36%)	
No	131 (65.8%)	51 (68.9%)	80 (64%)	
**Abscess**				**<0.001**
Yes	14 (7%)	13 (17.6%)	1 (0.8%)	
No	185 (93%)	61 (82.4%)	124 (99.2%)	
**Peritonitis**				**<0.001**
Yes	14 (7%)	12 (16.2%)	2 (1.6%)	
No	185 (93%)	62 (83.8%)	123 (98.4%)	
**WBC count**	15.08 (4.96)	16.49 (5.47)	14.25 (4.46)	**0.010**
**NEUT (%)**	77.88 (11.6)	81.07 (9.94)	75.99 (12.07)	**<0.001**
**PLT**	310.54 (83.81)	332.57 (98.13)	297.50 (71.32)	**0.032**
**CRP (mg/dL)**	4.7 (6.28)	8.11 (8.41)	2.66 (3.20)	**<0.001**
**RLQ tenderness to percussion, coughing, hopping**				**<0.001**
Yes	137 (68.8%)	74 (100%)	63 (50.4%)	
No	62 (31.2%)	0 (0)	62 (49.6%)	
**Anorexia**				**0.023**
Yes	125 (62.8%)	54 (73%)	71 (56.8%)	
No	74 (37.2%)	20 (27%)	54 (43.2%)	
**Body temperature ≥38 °C**				**0.017**
Yes	68 (34.2%)	33 (44.6%)	35 (28%)	
No	131 (65.8%)	41 (55.4%)	90 (72%)	
**Nausea/vomiting**				**0.038**
Yes	63 (31.7%)	30 (40.5%)	33 (26.4%)	
No	136 (68.3%)	44 (59.5%)	92 (73.6%)	
**Tenderness over RIF**				-
Yes	199 (100%)	74 (100%)	125 (100%)	
**Pain migration to RLQ**				**<0.001**
Yes	58 (29.1%)	34 (45.9%)	24 (19.2%)	
No	141 (70.9%)	40 (54.1%)	101 (80.8%)	
**Duration of symptoms (h)**				0.671
<24	68 (34.2%)	28 (37.8%)	40 (32%)	
24–48	86 (43.2%)	31 (41.9%)	55 (44%)	
>48	45 (22.6%)	15 (20.3%)	30 (24%)	

**Table 2 diagnostics-13-02275-t002:** Binary logistic regression results for acute complicated appendicitis.

	Univariate Logistic Regression		Multivariate Logistic Regression	
	OR (95% CI)	*p*	OR (95% CI)	*p*
Appendiceal diameter ≥ 8.45: Yes	3.79 (2.04–7.03)	<0.001	4.43 (1.46–13.45)	**0.009**
Anatomical position: Usual	2.57 (0.82–8.00)	0.104	4.85 (0.80–29.43)	0.086
Distinct appendiceal wall layers: Yes	3.02 (1.67–5.49)	<0.001	0.87 (0.29–2.58)	0.799
Non-compressible: Yes	1.52 (0.46–5.04)	0.492	1.40 (0.15–12.64)	0.765
Target sign appearance: No	3.30 (1.81–6.01	<0.001	4.12 (1.39–12.18)	**0.010**
Hypervascularisation: Yes	0.23 (0.03–1.92)	0.174	0.80 (0.06–10.17)	0.865
Appendicolith: Yes	4.58 (2.30–9.12)	<0.001	6.50 (2.21–19.16)	**0.001**
Periappendiceal fat inflammation: Yes	2.74 (1.47–5.11)	0.001	1.40 (0.47–4.19)	0.551
Free abdominal fluid: Yes	1.86 (1.01–3.48)	0.051	2.12 (0.39–11.56)	0.385
DFIF: Yes	2.39 (1.05–5.45)	0.037	2.50 (0.53–11.75)	0.247
PFIF: Yes	1.88 (1.05–3.37)	0.035	0.90 (0.20–4.04)	0.893
DPFIF: Yes	0.71 (0.35–1.42)	0.330	0.26 (0.06–1.07)	0.062
Lymphadenitis: Yes	0.80 (0.43–1.48)	0.480	1.83 (0.64–5.22)	0.258
Abscess: Yes	26.43 (3.38–206.7)	0.002	40.05 (2.21–724.56)	**0.012**
Peritonitis: Yes	11.9 (2.58–54.85)	0.001	29.27 (1.03–832.36)	**0.048**
WBC ≥ 15.96: Yes	1.69 (0.95–3.03)	0.076	0.55 (0.17–1.74)	0.305
NEUT ≥ 78.95: Yes	2.48 (1.35–4.56)	0.003	3.48 (1.11–10.88)	**0.032**
PLT ≥ 321.5: Yes	2.09 (1.16–3.75)	0.014	2.32 (0.84–6.42)	0.106
CRP ≥ 1.99: Yes	4.55 (2.38–8.70)	<0.001	3.46 (1.23–9.71)	**0.018**
Anorexia: Yes	2.05 (1.10–3.83)	0.024	3.02 (0.97–9.34)	0.056
Body temperature ≥ 38 °C: Yes	2.07 (1.13–3.78)	0.018	3.15 (1.07–9.29)	**0.038**
Nausea/vomiting: Yes	1.90 (1.03–3.50)	0.039	0.93 (0.32–2.70)	0.895
Pain migration to RLQ: Yes	3.58 (1.89–6.77)	<0.001	4.17 (1.44–12.12)	**0.009**
Duration of symptoms (h): <24	1.40 (0.64–3.07)	0.401	7.76 (1.48–40.60)	**0.015**
Duration of symptoms (h): 24–48	1.13 (0.53–2.41)	0.757	2.33 (0.52–10.51)	0.271

Note: Analyses are adjusted for sex and age, OR = Odds Ratio, CI = Confidence Interval, *p* < 0.05.

## Data Availability

The datasets generated during and/or analyzed during the current study are available from the corresponding author on reasonable request.

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
