# Peer review of "Predicting Complicated Appendicitis in Children: Pros and Cons of a New Score Combining Clinical Signs, Laboratory Values, and Ultrasound Images (CLU Score)"

_diagnostics, 2023, doi:10.3390/diagnostics13132275_

Round 1
Reviewer 1 Report
When reading this paper the authors mention new instruments. I think they need to look back in the history. Misses a couple of important early papers
#1 Puylart JBCM acute appendicitis-ultrasonography New Engl J Med 1987;317
#2 Eriksson S Laboratory tests -suspected appendicitis Acta Chir Scand 1989:155
The first described graded compression the second described for the first time combined testing CRP-WCC-Neutr .
There are other papers missing :
#3 Fawkner-Corbett D Diagnostic blood tests in paedriatric app.
BMJ Open 2022;12
67 studies are analysed, which should be mentioned
#4 Ohmann C 10 scoring systems evaluated Eur J Surg 1995, 161
None were good, why should this be any better?
#5 Gans et al Guideline for the diagnostic pathway in pat AAP Dig Surgery 2015; 32
Should be mentioned- no scoring system, no imaging is perfect.
#6 Eriksson Diagnostic value of repetitive analyses in susp AA Scand J Gastroenterol 1994:29
It should be comment since it is well known that CRP rises and WCC decline
before operation.
I believe in rewrite the introduction, discussion and take in these references in paper.
there is a couple of misspelling that should be corrected
Author Response
Thank you very much for your valuable comments.
We think that in this new version we tried to include all your highlights.
First of all, we didn’t use any new instrument, but we tried to combine as much parameters as possible in order to have a more credible score for the diagnosis not only of the acute appendicitis but the complicated one in children, trying to help clinicians to avoid negative (unnecessary) appendectomies.
We changed according to your advice as much as we could the introduction and the discussion and we added all the references you proposed. All the changes we made are marked in red inside the manuscript. We tried also to fix some misspelling we observed.
Reviewer 2 Report
No comments.
Author Response
Thank you very much for your acceptance.
Reviewer 3 Report
well done and interesting paper to read and to know about
Author Response

(The authors gave the same response as above.)

Reviewer 4 Report
The Authors carried out the study titled “Predicting complicated appendicitis in children. A new score after combination of Clinical signs, Laboratory values and Ultrasound images (CLU score). Pros and cons of the method”. The retrospective study aimed to combine clinical signs, laboratory values and ultrasound images to create a new score to distinguish preoperatively the uncomplicated of appendicitis from the complicated ones. The results are very interesting.
In my opinion the ultrasound (US) parameters are overbalanced versus clinical and laboratory parameters, since the US method is operator’s diagnostic ability. Furthemore the Authors could better to clarify:
- If it is possible to balance the parameters of the score;
- If it possible to describe and share the specific US protocol approach used for diagnosis of acute appendicitis; it may be helpful to apply the new score in different pediatric centers.
no
Author Response
Thank you very much for your valuable comments.
We think that in this new version we tried to include all your highlights.
To answer at your first question the answer from our statistical team is that: To create the CLU score, a value of 1 was assigned to each of the ten parameters because no high correlations were found between the variables and thus, they could be considered as single variables.
According to your second question
We added marked in red all the protocol of our team pediatric radiologist and also the reference of our previous published article that is referred analytically to US findings.
Round 2
Reviewer 1 Report
No Otter comments it is fine